# Optical Tissue Clearing to Study the Intra-Pulmonary Biodistribution of Intravenously Delivered Mesenchymal Stromal Cells and Their Interactions with Host Lung Cells

**DOI:** 10.3390/ijms232214171

**Published:** 2022-11-16

**Authors:** Alejandra Hernandez Pichardo, Francesco Amadeo, Bettina Wilm, Raphaël Lévy, Lorenzo Ressel, Patricia Murray, Violaine Sée

**Affiliations:** 1Department of Molecular Physiology and Cell Signalling, Faculty of Health and Life Sciences, University of Liverpool, Liverpool L69 3BX, UK; 2Centre for Preclinical Imaging, Faculty of Health and Life Sciences, University of Liverpool, Liverpool L69 3BX, UK; 3INSERM, LVTS, Université Sorbonne Paris Nord, F-75018 Paris, France; 4Department of Veterinary Anatomy Physiology and Pathology, Faculty of Health and Life Sciences, University of Liverpool, Liverpool L69 3BX, UK; 5CNRS UMR 5305, Tissue Biology and Therapeutic Engineering Laboratory (LBTI), University Claude Bernard Lyon1, 69007 Lyon, France

**Keywords:** optical tissue clearing, mesenchymal stromal cells, mouse lung, biodistribution, cell tracking

## Abstract

Mesenchymal stromal cells (MSCs) injected intravenously are trapped in the capillaries of the lungs and die within the first 24 h. Studying the biodistribution and fate of labelled therapeutic cells in the 3D pulmonary context is important to understand their function in this organ and gain insights into their mechanisms of action. Optical tissue clearing enables volumetric cell tracking at single-cell resolution. Thus, we compared three optical tissue-clearing protocols (Clear, Unobstructed Brain/Body Imaging Cocktails and Computational analysis (CUBIC), modified stabilised 3D imaging of solvent-cleared organs (s-DISCO) and ethyl cinnamate (ECi)) to evaluate their potential to track the biodistribution of human umbilical cord MSCs expressing the tdTomato fluorescence reporter and investigate how they interact with host cells in the mouse lung. The results showed that although CUBIC clearing is the only method that enables direct imaging of fluorescently labelled MSCs, combining s-DISCO or ECi with immunofluorescence or dye labelling allows the interaction of MSCs with endothelial and immune cells to be studied. Overall, this comparative study offers guidance on selecting an optical tissue-clearing method for cell tracking applications.

## 1. Introduction

Cell-based therapies include the administration of exogenous cells to trigger a regenerative response. Several sources of therapeutic cells have been investigated [1]. The majority of clinical trials have explored the potential of mesenchymal stromal cells (MSCs). Although it has been reported that MSCs are multipotent, their therapeutic effects in vivo are mostly mediated by secreted factors that promote the repair of injured host tissues and modulate the host’s immune system [2]. Increasing evidence suggests that the intravenous administration of MSCs and various other cell types is followed by the entrapment of most of the administered cells in the lung capillaries. Little is known about the role of therapeutic cells in this organ as cell persistence is low, with most cells dying within the first 24 h post systemic administration [3].

One of the most common safety issues following the intravascular infusion of MSC therapies is thromboembolic complications. Thus, it is important to analyse the distribution of the MSCs within the vasculature to determine their potential to occlude the pulmonary vessels [4]. Moreover, MSCs can exert their therapeutic effects via immune effector cell mediation triggered by MSC death in the lungs [5], but the interactions of the MSCs with different immune cell populations in the lung remains to be established. Cell tracking by following the biodistribution of labelled therapeutic cells within tissues might offer insights into these questions.

Cell tracking in tissues at single-cell resolution has been traditionally conducted using thin-section histological analyses. Although useful, the field of view is limited and might not be an accurate representation of the whole organ [6]. Investigating the biodistribution of cells in thick tissue sections offers an advantage to thin-section analyses, but biological tissues are dense and inherently scatter light, preventing the visualisation of deeper structures [7]. Optical tissue clearing is a technique that minimises the heterogeneities within tissues by removing lipids and matching the refractive index (RI) between the sample and the imaging medium [8]. As a result, opaque samples become more transparent, and the decrease in light scattering allows deep-tissue imaging [9].

Optical tissue clearing, in combination with molecular labelling and optical sectioning microscopy, has become an important tool for 3D imaging in several biological applications, including investigating the biodistribution of cells in whole organs.

Despite the many advantages of optical tissue clearing, practical limitations to the applicability of clearing protocols for imaging remain. Several protocols have been developed in recent years [10], and selecting between them requires the careful consideration of a range of parameters to achieve the optimal trade-off for specific applications. Sample size and tissue composition impact the clearing speed and limit the microscopes that can be used when imaging large samples [11]. Fluorophore preservation poses another challenge, as certain clearing protocols are incompatible with fluorescent probes [12]. In particular, the preservation of protein-based fluorescence and lipid staining remain open challenges in the field [8]. The compatibility of immunostaining with the chemicals used for clearing as well as antibody penetration in large samples requires testing and optimisation [11]. Moreover, certain parameters differ between tissue and sample types, and no single clearing approach fits all, necessitating the use of application/tissue-specific protocols.

Several studies have focused on clearing lung tissue to investigate biological processes. An example is the study of blood vessel formation, achieved by labelling the intact vasculature with the organic dye Evans Blue [13]. Nevertheless, no study to date has performed optical tissue clearing to investigate the biodistribution of administered MSCs within the lungs. Here, we compared three different tissue clearing protocols: CUBIC, a modified s-DISCO and ECi. CUBIC was chosen based on its ability to preserve fluorescent proteins [14]; ECi due to its cost-effectiveness, safety and ease of access to the reagents required [15]; and s-DISCO due to its reported compatibility with fluorescent proteins despite being a solvent-based method [13].

The comparison of these methods was performed with the aim of imaging thick lung slices in 3D to track MSCs labelled with the genetic reporter tdTomato and study their biodistribution within the host’s lung. Moreover, as a proof of principle, we explored the possibility of using optical tissue clearing to investigate the interactions of the administered MSCs with the host’s endothelial cells and immune microenvironment.

## 2. Results

### 2.1. Bioluminescence Imaging Reveals hUC-MSC Entrapment in the Lungs

To monitor the fate of MSCs administered in mice, the cells were stably labelled with fluorescent and luminescent reporters. The tdTomato reporter was used as it is the brightest red shifted fluorescent protein available, helping overcome the high autofluorescence of tissues (Figure 1a, left) [16]. In vitro, the level of emitted bioluminescence is dependent on the number of hUC-MSCs present (Figure 1a, right). Immediately after the IV administration of hUC-MSCs, their localisation was monitored in vivo using bioluminescence imaging, allowing us to observe that the hUC-MSCs were localised to the lungs (Figure 1b).

### 2.2. Lung Clearing Comparison after Different Optical Tissue-Clearing Methods

Whilst bioluminescence is useful for monitoring the whole body biodistribution of hUC-MSCs in vivo, the low spatial resolution precludes a detailed analysis of the cell distribution within the lung. To better understand why the cells were retained in the lungs, and to analyse their impact on the host at the molecular, cellular and tissue level, we sought to monitor cell biodistribution in the lung tissues ex vivo at single-cell resolution. To achieve this, we compared three different optical tissue-clearing methods that would enable the cells to be visualised in thick lung sections: CUBIC, a modified s-DISCO and ECi. s-DISCO and ECi are solvent-based methods that cleared whole mouse lungs in a matter of hours, while CUBIC uses water-based reagents and required several days to entirely clear tissues with an average lipid-clearing time of 3 days (Figure 2a).

To reach transparency, the samples were immersed in RI matching solutions, and representative images of the cleared lungs are shown in Figure 2b. We quantified the resulting levels of transparency from digital images taken of the tissues before and after clearing. The CUBIC protocol resulted in the highest transparency of the lungs, reaching close to 90%, while s-DISCO and ECi demonstrated transparency of approximately 30% (Figure 2c). Additionally, we calculated the size change and found that there was a significant increase in size after CUBIC clearing. The opposite was observed after s-DISCO and ECi clearing, where the samples shrunk (Figure 2d). We observed the presence of adequate air spaces, bronchioles, alveolar sacs, and blood vessels in images acquired by recording tissue autofluorescence. Overall, this indicates that the characteristic lung structures remain detectable after all the clearing protocols (Figure 2e).

### 2.3. Effect of Different Optical Tissue-Clearing Methods on the Preservation of Fluorescence in the Lungs

The ability to detect fluorescent dyes and fluorescent proteins is essential to study the environment and fate of the injected cells, yet fluorescence quenching is one of the key limitations of many organic solvent-based optical tissue-clearing methods. Therefore, we compared the effect of CUBIC, modified s-DISCO and ECi on the tdTomato fluorescent label intensity of the hUC-MSCs. By acquiring fluorescent images of cleared 500 µm lung sections immediately after clearing and 3 days after the samples had been stored in RI matching medium, we found that CUBIC preserved tdTomato fluorescence upon all clearing steps and that storage did not affect the fluorescence of tdTomato. On the other hand, the solvent-based methods increased background autofluorescence and appeared to quench the tdTomato fluorescence, suggesting that these are unsuitable methods to detect this fluorescent protein (Figure 3a). Interestingly, clearing thinner lung sections (100 µm) using a reduced dehydration time (Appendix A) allowed for the detection of tdTomato immediately after both modified s-DISCO and ECi, indicating that dehydration time is a key parameter in preserving the fluorescence of proteins. Moreover, the storage of the thin sections in ECi for 3 days did not result in tdTomato quenching, but storage in dibenzyl ether (DBE) in the modified s-DISCO protocol did (Appendix A).

Systemic cell administration results in a large fraction of the cells becoming entrapped in the lung’s vasculature [4]. Since Evans Blue allows the labelling of the vasculature to analyse IV-administered hUC-MSCs in the context of the pulmonary 3D vascular network, we determined whether Evans Blue labelling was affected by the clearing protocols. Our analysis showed that Evans Blue is incompatible with CUBIC since all fluorescent signals were lost (Figure 3b). By contrast, Evans Blue fluorescence was not only preserved by s-DISCO and ECi, but the fluorescence intensity of the dye had increased after clearing with either solvent-based method. We speculate that the shrinkage of the samples leads to a higher density of fluorescent molecules and subsequent increase in signal intensity, since the samples were imaged under the same conditions.

### 2.4. Compatibility of Antibody Labelling with Different Clearing Methods

Although the tdTomato fluorescence of hUC-MSCs was not detectable in s-DISCO- and ECi-treated thick lung sections, the biodistribution of administered cells can be investigated using cell-specific antibodies. To test the compatibility of each optical clearing method with antibody staining, we aimed to detect the hUC-MSCs with a specific antibody within the lungs. In this instance, we utilised the human origin of the hUC-MSCs and applied an antibody specific for human mitochondria to distinguish the hUC-MSCs from the mouse tissue. Alternatively, antibodies against the tdTomato or FLuc reporters could be applied.

Our image analysis showed that the human mitochondria antibody colocalises with the tdTomato signal in CUBIC-cleared samples, confirming the specificity of this antibody to the human hUC-MSCs (Figure 4a). Moreover, we observed that the antibody signal was detected throughout the 500 µm thick sample. Similarly, in the s-DISCO- and ECi-cleared samples, the antibody permeated the entire tissue (Figure 4b,c).

### 2.5. In Vivo and Ex Vivo Tracking of Administered hUC-MSCs by BLI and Optical Tissue Clearing

To evaluate the biodistribution and fate of hUC-MSCs in vivo, the animals were imaged using BLI immediately after the IV administration of the cells, and 24 h post injection. A strong signal was detected in the lungs on the administration day but was significantly reduced on day 1 (Figure 5a,b). We dissected whole lung lobes on the day of cell injection and 24 h post cell administration and performed CUBIC clearing, as this was the method that allowed the direct detection of the hUC-MSCs. Subsequent confocal imaging of lung lobes revealed that the cells distributed throughout the tissue and did not home preferentially to any site, as they could be found dispersed within the lung 2 h after administration (Figure 5c, left; Appendix A). Ex vivo imaging of CUBIC-cleared lungs collected 24 h after cell administration confirmed that most of the cells had been cleared from the lungs, as shown by the reduction in the size of the cell clumps (Figure 5c, right). The overall cell distribution remained similar as on the injection day, with the hUC-MSCs localising evenly throughout the organ (Figure 5c, right; Appendix A).

Given the compatibility of the CUBIC protocol with immunofluorescence, as proof of concept, we explored the possibility of using this clearing method to study the interaction of the hUC-MSCs with cells of the mouse lung. The vasculature was labelled with the CD31 endothelial marker, which showed that the hUC-MSCs appeared to be retained in the pulmonary microvasculature as no cells were detected within large blood vessels (Figure 5d) at 2 h (Appendix A) or 24 h (Appendix A).

Alternatively, the vasculature can be labelled by injecting Evans Blue IV. In the experiment described previously (Figure 4), the Alexa Fluor^®^ 647 secondary antibody was used, but its spectrum overlaps with Evans Blue. Due to an increase in tissue autofluorescence across all wavelengths, following ECi, we were unable to label the hUC-MSCs utilising the green, red or near-infrared channels. Thus, it was not possible to immunolabel the hUC-MSCs in lungs stained with Evans Blue and cleared by ECi. Nevertheless, the thin-section analysis of uncleared lungs stained with Evans Blue, by injecting dye IV after cell injection, revealed that hUC-MSCs remained in close contact with the pulmonary vasculature when the lungs were harvested immediately after cell administration (Appendix A). In addition, hUC-MSCs blocked the free flow of Evans Blue dye, as evidenced by the accumulation of dye around areas where the hUC-MSCs were present. Moreover, the lack of Evans Blue vascular staining in lung regions surrounded by hUC-MSCs suggested that the cells might have formed emboli [17] (Appendix A).

Finally, to demonstrate the usefulness of the CUBIC clearing protocol to study immune responses in the lung after hUC-MSC administration, neutrophils were stained with the myeloperoxidase (MPO) marker. Rapid neutrophil infiltration was observed 2 h after IV hUC-MSC injection with decreasing neutrophil levels after 24 h (Figure 5e).

In summary, of the three methods compared, the CUBIC protocol proved to be suitable for efficiently clearing the lung specimens without altering the tissue morphology. CUBIC was the only method that preserved tdTomato fluorescence in thick lung sections, allowing for the direct visualisation of hUC-MSCs by confocal microscopy. However, CUBIC failed to preserve the Evans Blue labelling of the vasculature but showed good antibody compatibility. In contrast, s-DISCO and ECi allowed the rapid optical clearing of whole lungs but permanently quenched the fluorescence of tdTomato, while preserving the endothelial Evans Blue signal. Despite the fact that no clearing method was perfect for all subsequent imaging applications, by combining the CUBIC clearing method with various staining approaches, we were able to demonstrate that hUC-MSCs were trapped in the lung microvasculature just after injection, potentially blocking blood flow, and that hUC-MSC intravenous administration triggers neutrophil infiltration.

## 3. Discussion

In this work, CUBIC, modified s-DISCO and ECi optical tissue clearing protocols were compared with the aim of establishing a suitable approach to study the biodistribution of hUC-MSCs in mouse lungs following systemic cell delivery.

Optical tissue clearing matches the RI of heterogeneous samples and reduces light scattering, enabling the investigation of biological processes in a 3D organ context. Broadly, clearing methods can be classified into water-based (hydrophilic) and solvent-based (hydrophobic) methods depending on the chemistry used [10]. The selection of a clearing protocol depends on parameters such as the size of the sample, tissue composition and the intended goal of the experiment.

First, it is relevant to characterise the effect of the clearing method on the tissue of interest. When applied to lung tissue, CUBIC is the superior method regarding tissue transparency, albeit taking 5 days to complete. In contrast, both solvent-based methods cleared the samples within hours but resulted in transparency below 50%. Changes in sample size occurred as expected: hydrophilic methods led to sample expansion, which might be an advantage when one is interested in increasing imaging resolution; solvent-based protocols resulted in sample shrinkage, which can be advantageous when imaging large samples [18].

Another relevant consideration when selecting a clearing protocol is whether there is a need to preserve fluorescent proteins. In this study, the red fluorescent protein tdTomato was used as a molecular label for the hUC-MSCs. tdTomato was immediately quenched after the s-DISCO and ECi clearing of thick lung sections, rendering these protocols unsuitable for the direct visualisation of fluorescently labelled cells, while CUBIC preserved the fluorescence of tdTomato.

Fluorescent proteins are stabilised by water molecules, and thus, dehydration results in their denaturation and subsequent loss of fluorescence [19]. To overcome this, the use of milder dehydrating chemicals such as tetrahydrofuran (THF) in the 3DISCO protocol [20], tert-butanol and diphenyl ether in the uDISCO protocol [21] and 1-propanol in the second-generation ECi protocol have been used, successfully allowing GFP preservation for days to months [15]. Given these reports, we used 1-propanol as the dehydration agent for both s-DISCO and ECi protocols in the hope of preserving tdTomato. Nevertheless, we were not able to detect tdTomato fluorescence. Interestingly, Glaser and colleagues acquired a 3D image of a whole mouse lung cleared with ECi via autofluorescence at 561 nm, which is consistent with our observation that tissue autofluorescence at this wavelength is increased after immersion in ECi [22]. Moreover, the compatibility of a variety of fluorescent probes with ECi was tested, and tdTomato fluorescence was reported to be poor after clearing [15].

In this study, the s-DISCO protocol was chosen given that it suggests that adding the antioxidant propyl gallate to DBE prevents the formation of peroxides and aldehydes, which are partly responsible for fluorescence decline, and makes it possible to preserve tdTomato for up to a year. This protocol is complex and requires specific expertise to purify the dehydration and RI matching reagents and eliminate all peroxide and aldehyde contents before adding the propyl gallate to prevent their regeneration [23]. We aimed to test whether simplifying the protocol to make it more accessible to the wider research community, by preventing peroxide formation in DBE, would suffice to preserve the fluorescence of tdTomato. Nevertheless, we did not observe fluorescence preservation after this modified s-DISCO clearing, indicating that this approach is not sufficient to preserve fluorescent proteins and suggesting that pure chemicals might be necessary throughout the entire clearing protocol [23]. The purification protocol requires great care given that the chemicals used are explosive.

Interestingly, we observed that the fluorescence of tdTomato was preserved immediately after the completion of the modified s-DISCO and ECi protocols by decreasing the dehydration time when clearing 100 µm thin lung sections. This finding might explain the variability in the results between users of the same protocols for different applications and samples [24]. Moreover, it indicates that optimising the duration of the dehydration steps might be necessary when implementing a solvent-based optical tissue-clearing method where fluorescent proteins are involved. Decreasing the dehydration time might result in protein-based fluorescence preservation, but it would come at a cost regarding sample transparency. Moreover, dehydration is not the only factor that affects tdTomato, as evidenced by the preservation of fluorescence upon storage in ECi, but not in DBE, indicating that the RI matching solvent also plays a critical role in fluorescence quenching.

Temperature and pH have also been proven to play an important part in fluorescence retention, with alkaline pH and 4 °C being the optimal parameters that allow fluorescent protein preservation. The studies that showed this were conducted specifically to preserve GFP fluorescence. Although we followed these recommendations, we failed to visualise tdTomato fluorescence.

The preservation of other fluorescent compounds, such as fluorescently labelled antibodies and synthetic organic dyes, is another consideration when selecting a tissue-clearing method for a particular application. To study hUC-MSC distribution with spatial landmarks in the 3D lung context, the vasculature was labelled with Evans Blue. CUBIC washed away the dye, while the modified s-DISCO and ECi preserved Evans Blue fluorescence. Other approaches to label the vasculature such as lectin staining, perfusion-based fluorescent dextran staining and perfusion with Cy7-PEI dye were attempted. Lectin stains were not compatible with CUBIC, while perfusion with fluorescent dextran or Cy7-PEI did not stain the vasculature homogeneously and were thus not taken forward for testing in combination with optical tissue clearing (not shown).

The immunofluorescent staining of whole organs enables the study of biological processes at a single-cell resolution while maintaining tissue integrity as well as structure and cell localisation. Nevertheless, staining thick tissues poses challenges that need to be considered when opting for this method, such as non-specific binding, poor antibody diffusion, long incubation times, high background, and long working distance objectives [25]. Although we did not stain the whole lung, 500 µm thick lung sections provide a better representation of the biodistribution of the administered hUC-MSCs in comparison with traditional thin sections standard in histological studies, while saving costs and reducing the need for specialised microscopy equipment.

The field of optical tissue clearing is moving at a fast pace with new protocols becoming available. Here, we did not find a single protocol that allowed us to preserve the fluorescence of tdTomato, stain the vasculature and perform immunostaining simultaneously, highlighting the challenges of establishing an optical tissue-clearing method for a new application. Alternative protocols that could be tested in the future include Ce3D, OPTIClear and MACS. The Ce3D protocol has been tested for lung tissue and is compatible with the immunofluorescent staining of immune cells while preserving the fluorescence of reporter proteins [26]. OPTIClear preserves fluorescence proteins and is compatible with lipophilic labels and most fluorescent dyes. Although this method decreases the brightness of some dyes, is incompatible with some primary antibodies and long-term storage in the clearing solutions decreases fluorescence, it might be explored for the purpose of cell tracking in the lungs [27]. Finally, MACS has also been validated for lung tissue and is compatible with fluorescent proteins as well as with lipophilic dyes that have been used to stain vascular structures, and antibody labelling, making it a promising approach [28].

In summary, CUBIC is the only protocol that preserved tdTomato, but it resulted in Evans Blue washing out of the sample. The opposite occurred when using modified s-DISCO or ECi, as tdTomato was quenched, but Evans Blue was preserved. These results reflect some of the considerations for optimising a clearing method for a specific application. Nevertheless, using immunofluorescence for the endothelial marker CD31 to label the vasculature in CUBIC-cleared lung sections seemed to indicate that the hUC-MSCs localise within the microvessels.

Finally, we briefly explored the possibility of using CUBIC to study the immune response in the lung to the administration of hUC-MSCs. A rapid infiltration of neutrophils was observed 2 h post cell injection, with the number of these cells decreasing after 24 h. This proof of principle paves the way for studying other immune cell populations in thick lung sections in the context of cell therapies.

## 4. Materials and Methods

### 4.1. Cell Culture and Stable Cell Line Generation

Human-umbilical-cord-derived mesenchymal stromal cells (hUC-MSCs) were obtained from the National Health Service Blood and Transplant (NHSBT, Liverpool, UK) at passage 3. The hUC-MSCs were transduced in the presence of 6 µg/mL DEAE-Dextran with a lentiviral vector pCDH-EF1-Luc2-P2A-tdTomato, encoding luc2 firefly luciferase (FLuc) reporter under the constitutive elongation factor 1-α (EF1α) promoter and upstream of a P2A linker followed by the tdTomato fluorescent protein (gift from Kazuhiro Oka; Addgene plasmid # 72486; http://n2t.net/addgene:72486; accessed on 10 November 2022; RRID:Addgene_72486). To obtain a >98% transduced population, the cells were sorted based on tdTomato fluorescence (BD FACS Aria, Sacramento, CA, USA). Cells were cultured in α-MEM supplemented with 10% FBS at 37 °C and in 5% CO_2_ and passaged at 80% confluence.

### 4.2. Animal Experiments

Eight-to-ten-week-old female albino mice (C57BL/6) (B6N-TyrC-Brd/BrdCrCrl (n = 15), originally purchased from the Jackson Lab (Sacramento, CA, USA) were used for all animal experiments. Mice were housed in individually ventilated cages (IVCs) under a 12 h light/dark cycle and provided with standard food and water ad libitum. All animal procedures were performed under a license granted by the Home Office under the Animals (Scientific Procedures) Act 1986 [29] and were approved by the University of Liverpool Animal Welfare and Ethics Review Board. Mice received 2.5 × 10^5^ FLuc-tdTomato-hUC-MSCs (hUC-MSCs hereinafter) suspended in 100 μL of PBS by intravenous (IV) administration via the tail vein under inhaled anaesthesia with isoflurane.

### 4.3. Bioluminescence Imaging

In vitro bioluminescence was performed by seeding a range of cell densities (from 625 to 2 × 10^4^ cells/well) into an optical bottom 96-well plate with black walls (#165,305, ThermoFisher, Altrincham, UK). The cells were allowed to attach for 3 h prior to the addition of 5.12 mM D-Luciferin. Imaging was performed immediately after substrate addition without an emission filter, a 13.3 cm field of view (FOV), an f-stop of 1 and a binning of 8. For in vivo bioluminescence imaging, mice received a subcutaneous (SC) injection of D-Luciferin (10 μL/g (body weight) of a 47 mM stock solution) after cell injection. After 20 min, the animals were imaged with an IVIS Spectrum instrument (Perkin Elmer, Beaconsfield, UK). Data are displayed in radiance (photons/second/centimeter2/steradian), where the signal intensity scale is normalised to the acquisition conditions. Acquisition was performed without an emission filter, a 22.8 cm FOV, an f-stop of 1 and a binning of 8.

### 4.4. Tissue Preparation

Immediately after bioluminescence imaging (BLI), the animals received an IV injection of Evans Blue (Merck, Darmstadt, Germany; 3 µL/g) for vascular labelling. The dye was allowed to circulate for 5 min before proceeding with a retrograde perfusion fixation protocol [30]. The animals received an intraperitoneal overdose of pentobarbital (Pentoject, Animal Care, York, UK; 100 µL) followed by cannulation of the abdominal aorta, opening of the vena cava and flushing PBS with a manual pump at a constant pressure of 200 mbar (Appendix A) for 6 min, to remove all blood cells, followed by 6 min of perfusion with 4% paraformaldehyde (PFA) to fix the whole animal. The total volume of each solution used per animal was 40 mL. The trachea was tied tightly with a surgical suture before opening the thoracic cavity for lung dissection. Finally, the lungs were post fixed in 4% PFA overnight at 4 °C.

### 4.5. Optical Tissue Clearing

#### 4.5.1. Solvent-Based Tissue Clearing: Modified s-DISCO and ECi

The general procedure consists of dehydrating fixed samples by sequentially adding pH9-adjusted solvents chilled to 4 °C. After dehydration, the respective RI matching solution was added until the samples reached optimal transparency for 3D imaging. The solvents used and incubation duration are detailed in Table 1.

#### 4.5.2. Aqueous-Based Tissue Clearing: Clear, Unobstructed Brain/Body Imaging Cocktails and Computational Analysis (CUBIC)

The CUBIC-cancer protocol was followed [31]. Briefly, CUBIC-L2 solution (L2), for delipidation and decolourisation, was prepared as a mixture of 10 w%/10 w% Triton X-100 (Merck, Darmstadt, Germany)/N-buthyldiethanolamine (B0725 Tokyo Chemical Industry, Oxford, UK). CUBIC-R2 solution (R2), for RI matching (RI = 1.52), was prepared as a mixture of 30% (*w*/*v*) nicotinamide (Merck, Darmstadt, Germany) and 45% (*w*/*v*) antipyrine (Merck, Darmstadt, Germany).

For whole-organ clearing, 4% PFA fixed lungs were washed with PBS for 2 h, three times each, followed by immersion in CUBIC-L1 solution (L1) (50% (*v*/*v*) mixture of water and CUBIC-L2) for 6 h at 37 °C. Then, the organs were immersed in L2 solution at 37 °C for 48 h. The L2 solution was refreshed after 24 h during this process. After decolourisation and lipid clearing, the organs were washed with PBS at room temperature for 2 h, followed by immersion in CUBIC-R1 solution (R1) (50% (*v*/*v*) mixture of water and CUBIC-R2) for 6 h at room temperature. Finally, organs were immersed and stored in R2 solution at room temperature overnight.

### 4.6. Size Change and Transparency Measurements

Fixed adult mouse lungs were cleared and imaged before and after clearing. The size and transparency of the samples were outlined and calculated using ImageJ (by drawing the region of interest and measuring the mean pixel intensity) (NIH, Bethesda, MA, USA) [32]. The area (cm^2^) of each single lung lobe was calculated by delineation using the ROI tool, and the average of all samples was set as the before value for all comparisons. The median grey value of the cleared organ image was used to measure transparency by normalising the obtained value to the background of the same image [33].
$$Transparency=\frac{sample\;median\;grey\;value}{average\;background\;median\;grey\;value}\times 100$$

### 4.7. Immunofluorescence

The lungs were cryoprotected in 15% sucrose followed by 30% sucrose over a period of 48 h before being embedded in optimal cutting temperature (OCT) medium. The samples were cut into 500 µm sections on a cryostat (Thermo Fischer Scientific, Warrington, UK; Microm HM505E) at −20 °C and stored at −80 °C.

For CUBIC staining, the tissue sections were delipidated by immersion in 50% CUBIC-L for 30 min at 37 °C followed by overnight incubation in CUBIC-L at 37 °C with shaking. All sections were washed 3× with PBS for 5 min. Tissues were incubated with zenon Alexa Fluor^®^ 647 (Invitrogen, Loughborough, UK; Z25008) labelled human mitochondria primary antibody (Merck, Darmstadt, Germany; MAB1273), 1:500 for 48 h at 4 °C and washed with PBS overnight at 4 °C. For CD31 (R&D systems, Minnesota, US; AF3628) staining, the lung sections were blocked overnight with PBS-TxDBN buffer (1× PBS, 2% TritonX-100, ddH2O, 2% BSA, 20% DMSO) at 37 °C and incubated with CD31 1:100 antibody for 72 h at 37 °C. Upon an overnight washing step, secondary antibody (Alexa Fluor^®^ 647, Thermo Fischer Scientific, Warrington, UK) incubation was conducted at 37 °C for 24 h. After the final overnight washing step, solvent-based or CUBIC clearing was performed as indicated in the clearing section.

### 4.8. Imaging

Cells in culture were imaged by light microscopy with a Leica DM IL microscope coupled to a DFC420C camera. Confocal images were acquired on a Leica DMi8 with an Andor Dragonfly spinning disk, coupled to an EMCCD camera using a 10×/0.45 air objective. Z-stacks were captured using the 488, 561 and 637 nm laser lines. The emission filters used were 525/50, 600/50 and 700/75. Maximum-intensity projections, three-dimensional reconstructions and image analyses were completed using the IMARIS version 9.9.0 (Bitplane, Schlieren, Switzerland) software packages, processed with ImageJ 3D viewer version 1.53c [32]. Three-dimensional surfaces were obtained via un-stacking the .ims image in Image J and reconstructed via variable threshold intensity (3D slicer). Three-dimensional elaboration was performed using NVIDIA^®^ Quadro 6000 GPUs and exported as .stl.

### 4.9. Statistics

The GraphPad Prism software version 8.4.3 was used to conduct the statistical analysis. The mean and standard deviation are used to represent all values in graphs. The number of replicates included in the analyses, as well as the type of statistical test used, are given in the figure legends.

## 5. Conclusions

We compared three optical tissue-clearing methods to track the lung biodistribution of hUC-MSCs labelled with the fluorescent protein tdTomato. The direct detection of the tdTomato cells was only possible using the CUBIC clearing protocol, which, although time-consuming, resulted in highly transparent lungs and showed good antibody compatibility and penetration. Moreover, using immunofluorescent staining allows the study of the interaction of the hUC-MSCs with cells in the host’s lung. Using 3D imaging of CUBIC-cleared lungs, we showed that hUC-MSCs were trapped in the pulmonary microvasculature, triggering an innate immune response, and were mostly cleared within the first 24 h after IV injection.

## Figures and Tables

**Figure 1 ijms-23-14171-f001:**
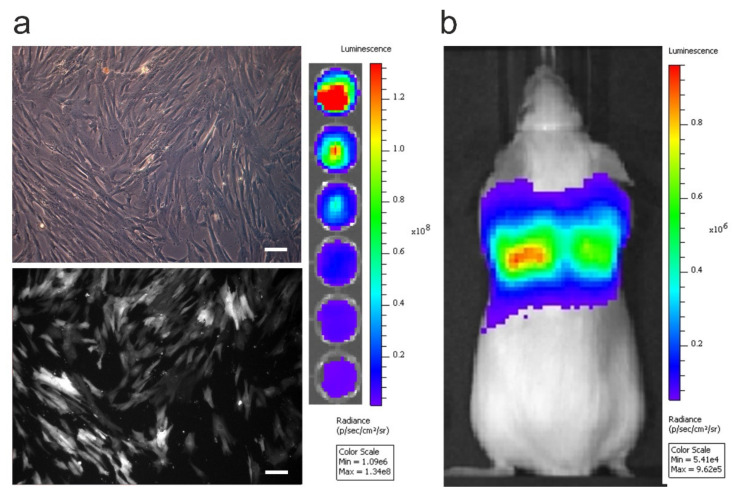
In vitro and in vivo imaging of hUC-MSCs. (**a**) Transmitted (top) and epifluorescence (bottom) images of transfected hUC-MSCs in culture. Scale bar = 100 µm (left). Representative image of hUC-MSCs seeded at decreasing concentrations (from 2 × 10^4^ to 625 cells/well) and treated with 5.12 mM D-Luciferin (right). (**b**) In vivo biodistribution of 2.5 × 10^5^ hUC-MSCs 20 min after intravenous administration.

**Figure 2 ijms-23-14171-f002:**
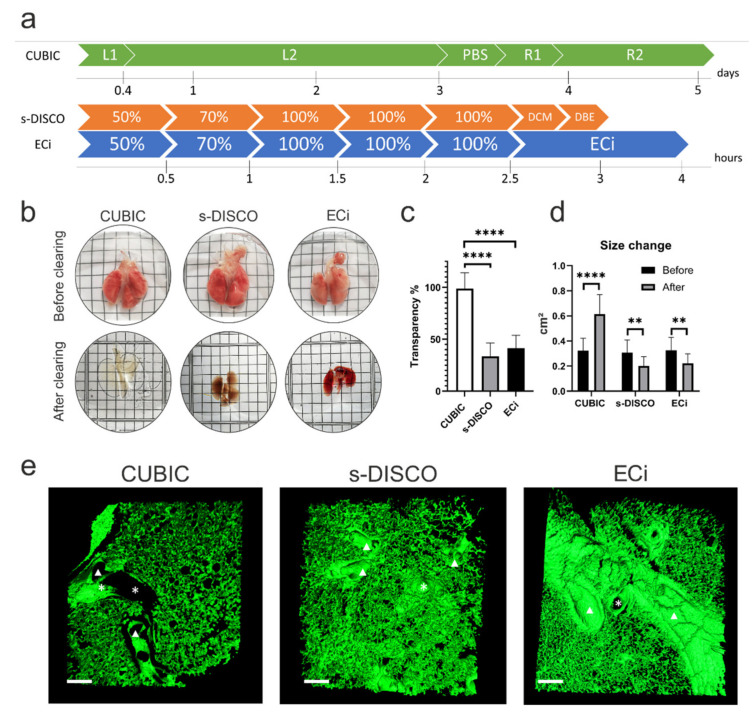
Comparison of the effect of different clearing methods on lung tissue. (**a**) Timeline required to clear whole lungs using each protocol. L1 = Delipidation and decolourisation cocktail 50% (*v*/*v*). L2 = Delipidation and decolourisation cocktail 100%. R1 = RI matching cocktail 50% (*v*/*v*). R2 = RI matching cocktail 100%. Percentages in s-DISCO and ECi represent the solvent concentration. (**b**) Representative images of lungs before (top row) and after (bottom row) optical tissue clearing. Each line of a square represents 2 mm. (**c**) Quantification of transparency of lung samples after optical tissue clearing One-way ANOVA with Tukey’s multiple comparisons test **** *p* < 0.0001. n = 3. (**d**) Size change (area cm^2^) in single lung lobes before and after each tissue-clearing method was evaluated using multiple paired *t*-tests ** *p* < 0.01, **** *p*  <  0.0001, n = 3. (**e**) Three-dimensional variable threshold intensity surface reconstruction of lung sections cleared by the different protocols. CUBIC-, s-DISCO- and ECi-cleared lungs show that the characteristic lung structures such as large blood vessels (indicated by asterisks (*)) and airways such as bronchi (indicated by arrowheads) are preserved after optical tissue clearing. Processing artefacts, due to tissue dehydration, are observed as cracks in the ECi-cleared sample. Scale bar = 150 µm. Three-dimensional MIPs before reconstruction can be viewed in Appendix A.

**Figure 3 ijms-23-14171-f003:**
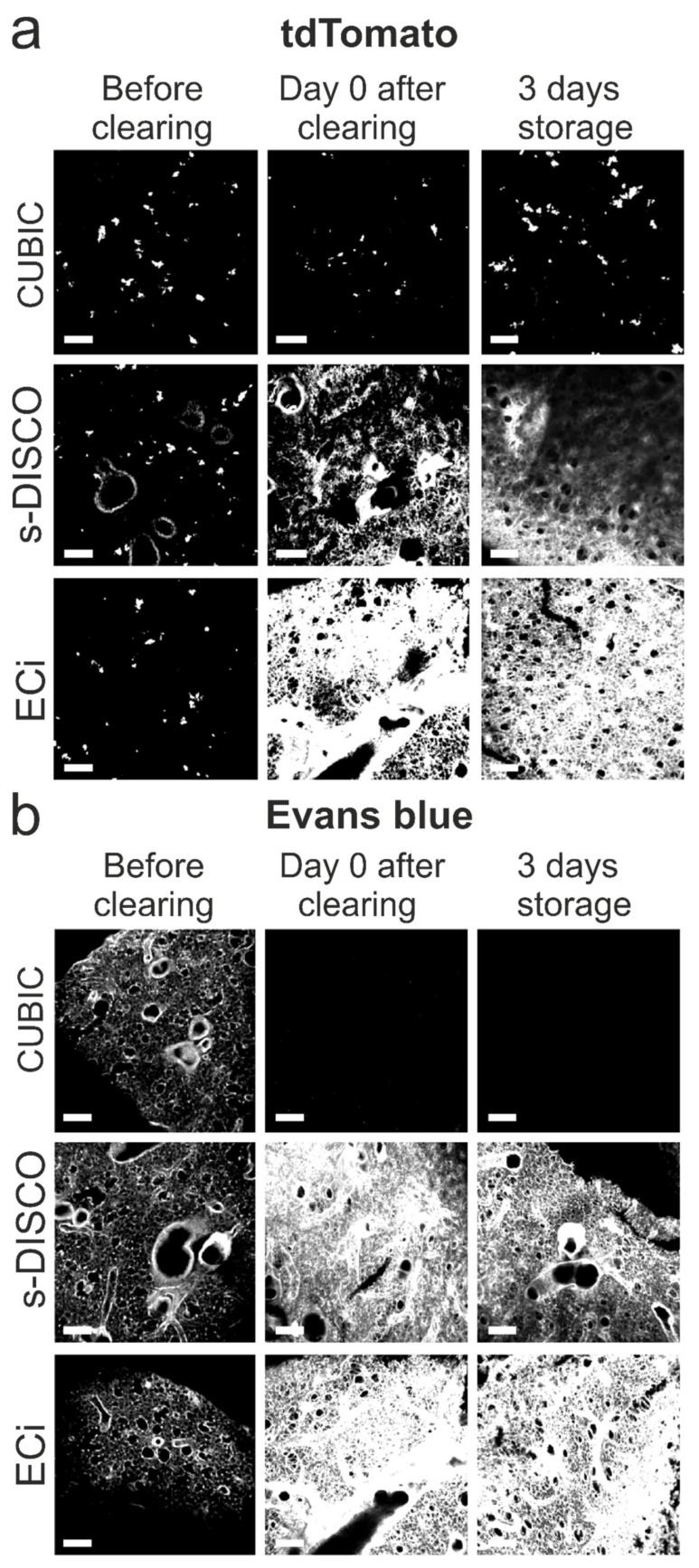
Fluorescence preservation of tdTomato and Evans Blue after clearing thick lung sections. Impact of CUBIC, s-DISCO and ECi clearing on the fluorescence of (**a**) tdTomato or (**b**) Evans Blue in 500 µm thick lung sections before, immediately after, and after storage for 3 days in RI solution. All confocal images are maximum-intensity projections (MIPs), taken from the surface of the slices. The same exposure times and display settings were used to acquire and visualise the images to reflect the change in the fluorescence preservation accurately. Scale bars = 150 µm.

**Figure 4 ijms-23-14171-f004:**
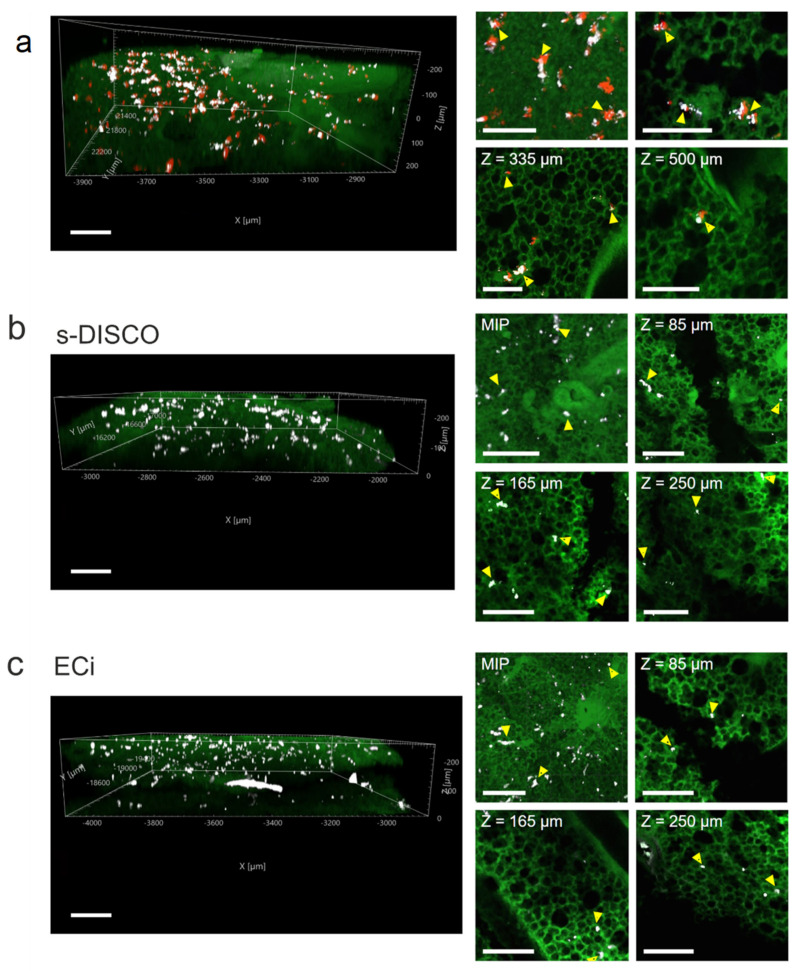
Preservation of immunofluorescence after optical tissue clearing. The human mitochondria antibody (white) was used to stain the tdTomato hUC-MSCs (red) in 500 µm lung sections followed by clearing with either CUBIC, s-DISCO or ECi. The antibody penetrated the entire depth in all samples. (**a**) Three-dimensional z-stack of a CUBIC-cleared lung section (left). Maximum-intensity projection and single slices at different sample depths (right). (**b**) Three-dimensional z-stack of an s-DISCO-cleared lung section (left). MIP and single slices at different sample depths (right). (**c**) Three-dimensional z-stack of an ECi-cleared lung section (left), MIP and single slices at different sample depths (right). hUC-MSCs are indicated by arrowheads. Scale bar = 200 µm (left), 150 µm (right).

**Figure 5 ijms-23-14171-f005:**
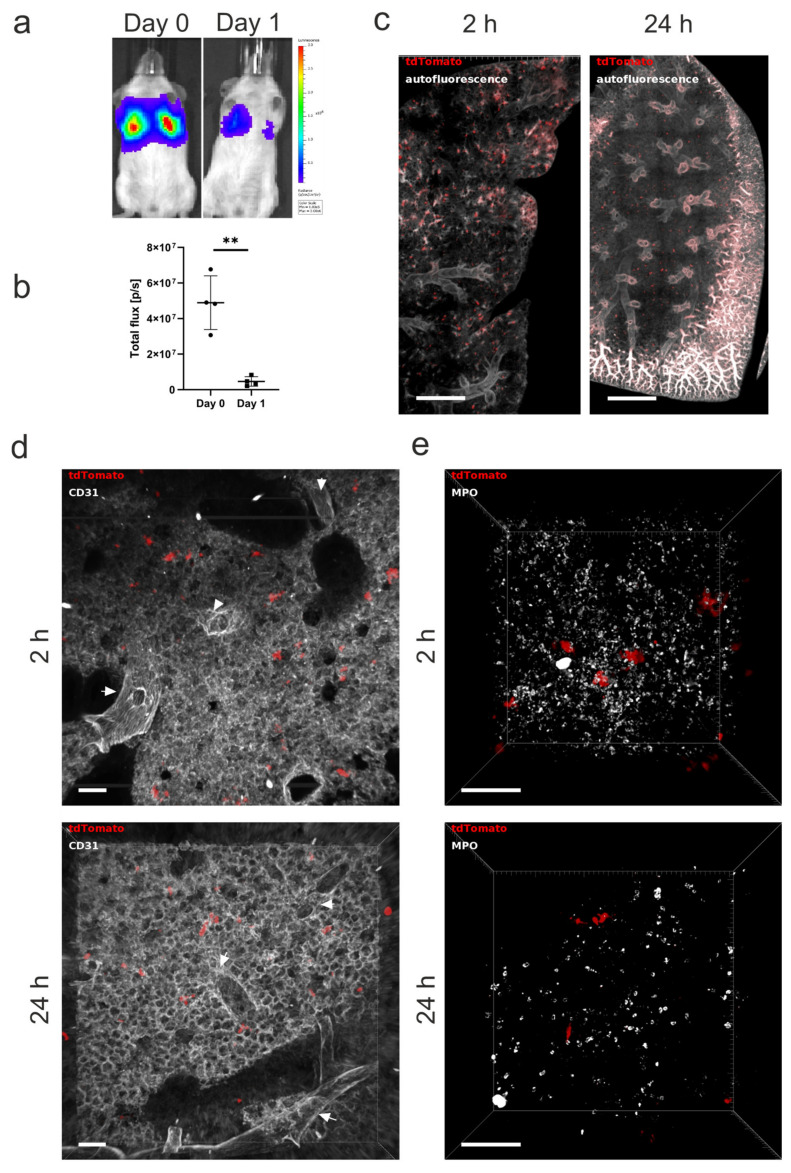
In vivo and ex vivo imaging to detect hUC-MSC distribution for up to 24 h post administration. (**a**) The 2.5 × 10^5^ hUC-MSCs were injected via the tail vein, and the mice were imaged on the administration day (Day 0) and 24 h post cell administration (Day 1). Representative images of the mice as acquired 20 min post subcutaneous administration of D-Luciferin. (**b**) Flux (light output) as a function of time. Data are displayed as mean ± SD from n = 4. Statistical analysis was performed using a paired student *t*-test. ** *p* < 0.05. (**c**) hUC-MSC biodistribution in whole mouse lung lobes after CUBIC clearing 2 h after cell administration and 24 h post injection. Scale bar = 800 µm. (**d**) CD31-stained 500 µm lung section on the day of administration and 24 h post cell injection. Large vessels are indicated by arrowheads. (**e**) Neutrophil recruitment to the lungs 2 and 24 h after hUC-MSC administration. MPO = myeloperoxidase. Scale bar = 100 µm.

**Table 1 ijms-23-14171-t001:** Solvent-based clearing protocols. The RI matching solvents used were ethyl cinnamate (ECi), Dichloromethane (DCM) and dibenzyl ether (dibenzyl ether (DBE)—(Merck, Darmstadt, Germany).

500 µm Sections			Whole Organ	
	s-DISCO	ECi	s-DISCO	ECi
1-propanol 50%	10 min	10 min	30 min	30 min
1-propanol 80%	10 min	10 min	30 min	30 min
1-propanol 100%	3 × 10 min	3 × 10 min	3 × 30 min	3 × 30 min
DCM	5 min	-	20 min	-
DBE w/0.4% propyl gallate	Storage	-	Storage	-
ECi	-	storage	-	storage

## Data Availability

The data that support the findings of this study are available to download from Zenodo at http://doi.org/10.5281/zenodo.6638775.

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
