# Peer review of "Optical Tissue Clearing to Study the Intra-Pulmonary Biodistribution of Intravenously Delivered Mesenchymal Stromal Cells and Their Interactions with Host Lung Cells"

_ijms, 2022, doi:10.3390/ijms232214171_

Round 1

Reviewer 1 Report

Here, in the manuscript entitled “Optical tissue clearing to study the intra-pulmonary biodistribution of intravenously delivered mesenchymal stromal cells and their interactions with host lung cells”, the authors aim to establish a suitable approach to study the biodistribution of hUC-MSCs in mouse lung by comparing three published clearing protocols, including CUBIC, sDISCO, and ECi. They concluded that CUBIC allowed direct imaging of fluorescently labeled MSCs, while sDISCO and ECi could not. There is no doubt that the demonstration of applying tissue clearing to the study of the intra-pulmonary biodistribution of MSCs provides a new perspective to researchers; the title of this manuscript also attracts me. However, in my opinion, the manuscript is poorly written, the figures are of poor quality or needless, and not convincing; hence they were hard to support the conclusions. It is a long way from being published in IJMS.

Here are some suggestions and questions for this manuscript.

1.      Figure 1a is not necessary at all.

2.      The lungs before clearing for each group (CUBIC, s-DISCO, ECi) are obviously different; for instance, the lung for CUBIC-group looks whiter than the other two, leading to the injustice of the comparison. Furthermore, the calculation method and description in Figure 2d are confusing: why the value of size change in the ‘Before’ group is between 0.2 and 0.4, there is no clear explanation in the text. More importantly, it is hard to conclude from Figure 2e that “tissue morphology remains unaffected by all the clearing protocols”. Additionally, the scale bars are lacking in Figure 2e.

3.      The quality of the images shown in Figure 3 is hard to read. What are the valid fluorescence signal in the grayscale Figure 3a and 3b?

4.      What does the red and white indicate in Figure 4? And the authors only stained 500μm-thick lung sections instead of the intact lung or lung lobe; it is far from enough to study the biodistribution of MSCs in lungs. The challenges of whole-mount immunostaining should be discussed.

5.      Figure 5c shows the biological distribution of HUC-MSC after tissue clearing. There should be more detailed descriptions for this result.

6.      This manuscript mentioned,’The vasculature was labelled with the CD31 endothelial marker, which showed that the hUC-MSCs appear to be retained in the pulmonary microvasculature as no cells were detected in the interstitium (Figure 5d).’ The images shown in Figure 5d can not support this conclusion.

7.      The authors tried to explain why Evans blue and antibody could not be used simultaneously in their work. Obviously, Evans blue is not an optimal choice in this case. There are multiple dyes that can be used to label the vasculature instead of Evans Blue. It is strongly suggested to consider another choice.

8.      In the original protocol of s-DISCO, s-DISCO can preserve tdTomato for up to a year. However, the author did not observe fluorescence preservation after s-DISCO clearing. However, in the original s-DISCO or FDISCO paper, tdTomato could be preserved well. The authors owed this inconsistency to the chemical purity. Actually, this problem is easy to be resolved by carrying out the experiments strictly according to the protocol of the original article. But the authors did not do that. In my opinion, they just failed this experiment.

9.      None of the three tissue clearing methods described in this manuscript are sufficient to complete the study, so why not use a different method?

10.   They should not use both ‘S-DISCO’ and ‘s-DISCO’ in the context.

Reviewer 2 Report

The main objective of this work is to investigate the possibility of using optical tissue clearing to undertand the interactions of the administered MSCs with the host’s endothelial cells and immune microenvironment.

The article is well described, it have good hypothesis, with adequate methodology, the cited references are mostly recent and relevant, and do not include self-citations. The manuscript’s results can be reproduced based on the methodology. The figures are clear and easy to understand.

Reviewer 3 Report

The manuscript by Pichardo et al. present a comparative study for selecting an optimal optical clearing method to investigate the MSC interaction with endothelial and immune cells within mouse lungs. Though similar comparative studies for tissue optical clearing have been published previously (Orlich et al, Histol. Histopathol., 2017, DOI: 10.14670/HH-11-903) (Xu et al, J. Biophotonics, 2019, DOI: 10.1002/jbio.201700187), the results for 3D biodistribution of hUC-MSCs in mouse lungs following systemic cell delivery presented in this study seems novel. The reviewer has several major concerns that need to be addressed.

1.       To acquire 3D distribution of fluorescent-labeled MSCs in mouse lung, the authors compared the performance of three tissue optical clearing methods. However, given that there are many excellent tissue clearing methods already established (e.g. Ce3D (Li et al, PNAS, 2017, DOI:10.1073/PNAS.1708981114), OPTIClear (Lai et al, Nat. Commun., 2018, DOI:10.1038/s41467-018-03359-w) and MACS (Zhu et al, Adv.Sci., 2020, DOI:10.1002/advs.201903185)), many of which may possess superior ability than the three methods selected in this study. Why did the authors choose these three methods for comparison? According to the results demonstrated, the authors seem not find a best method that totally meet their demands in the three methods. Therefore, the author may perform additional experiments for more clearing methods to ensure that their choice for clearing methods is optimal, or at least discuss.

2.       The authors displayed the clearing results for the three clearing methods and stated that CUBIC methods outperformed other two methods. However, the reviewer felt that the s-DISCO and ECi methods were not performed in an optimal manner. The time for dehydration each step is not enough for whole lungs, leading to insufficient transparency. The authors may prolonged the dehydration time from 0.5 h to 2-4 h, and the clearing performance would be better.

3.       The image quality for immunofluorescence in Figure 4 is not sufficient. The reviewer could not judge the compatibility of these methods with immunolabeling based on such results.

4.       As for a comparative study, quantitative results in this manuscript is lack. For example, quantitative comparison of fluorescence preservation in Figure 3 should be added. Statistical analysis of 3D hUC-MSC distribution in Figure 5 should be performed.

5.       The imaging depth in Figure 4 seem to be ‘um’ but not ‘nm’. Scalebars are missed in Figure 2b, 2e

Round 2

Reviewer 1 Report

 The authors have addressed part of my concerns but not my major concerns.

My major concerns are the novelty and whether the data shown support the conclusion.

As I mentioned in the 1st comment, none of the three tissue clearing methods used in the manuscript are sufficient to complete the study. The response letter from the authors did mention some other protocols,  Ce3D, OPTIClear and some other methods, which are of course the alternative protocols, but  I think they should be considered in this work but not be tested in the future since they are obviously not "new methods since starting this project". In my opinion, selecting suitable clearing methods should be the premise of the total work. Moreover, if the authors did not aim to image the whole lung but only lung slices, there are obviously better choices, such as the classical SeeDB, ScaleSQ, etc.

As for the sDISCO, it is unfair to give such results by changing its original protocol. The authors claimed that this protocol was complex (actually not), they changed the dehydration agent from THF to 1-propanol, and did not purify the DBE. However, the importance of purification of the agents had been emphasized in the original sDISCO paper, obviously, the authors chose a simple but not effective method in their study. To me, this is not rigorous and not scientific. Adding some descriptions in the manuscript can not address this issue. 

As I mentioned, Evans blues is not an optimal choice. The authors mentioned that they had tested Dextran and Cy7-PEI, but got negative results. However, some other probes and protocols, such as commonly used lectins and perfusion-based labeling protocols, are not even mentioned in the context.

Regarding the image quality, I think the quality is still too low and prone to misinterpretation. Taking Figure 3 as example, my question is what the valid signal in the grayscale images is. I think the readers can not distinguish the tdTomato signal from the autofluorescence as they are all white. The authors concluded from this figure that s-DISCO and ECi could quench tdTomato; the problem is that if you could not distinguish them, how could you conclude that? To be more detailed, as shown in Figure 3a, the image of the ECi-treated sample looks brighter; if you can not distinguish the signal and autofluorescence, it is hard to convince the readers about the conclusion. Here  I am emphasizing the image quality, the demonstration of the images are critical. 

Finally, I think each reviewer has his/her own concerns and opinions. I  DO NOT think I have to follow the others. I think that's why the journals have multiple reviewers for each manuscript.

I hope these comments and suggestions are helpful.

Reviewer 3 Report

The reviewer has no additional concerns about this manuscript.

Author Response

We thank the reviewer for their positive comments.